# Unraveling the Longitudinal Relationships Among Parenting Stress, Preschoolers’ Problem Behavior, and Risk of Learning Disorder

**DOI:** 10.3390/bs15060785

**Published:** 2025-06-06

**Authors:** Jie Huang, Dongqing Yu, Xiaoxue Tang, Yili Xu, Xiao Zhong, Xiaoqian Lai

**Affiliations:** 1Faculty of Education, Northeast Normal University, Changchun 130024, China; huangjie2318@126.com (J.H.); yudq2024@163.com (D.Y.); 15999340420@163.com (X.T.); 2Faculty of Education, Wenzhou University, Wenzhou 325035, China; xuyili2022@163.com; 3School of Education Science, Nanjing Normal University, Nanjing 210023, China; zxbegin1@163.com; 4School of Education Science, Baicheng Normal University, Baicheng 137000, China; 5Children and Childhood Education Research Center, Baicheng Normal University, Baicheng 137000, China

**Keywords:** parenting stress, problem behavior, risk of learning disorder, longitudinal study, preschool children

## Abstract

Problem behaviors and the risk of learning disorders in early childhood carry significant implications for children’s future development. Understanding the relationship between parenting stress and these developmental outcomes may inform the design of effective interventions to promote healthy child development. The present longitudinal study investigated the association between parenting stress and preschoolers’ risk of learning disorders, specifically focusing on the mediating role of problem behaviors. Data were collected at two time points, approximately 6 months apart, from 284 preschool-aged children and their parents (mean age of children at Time 1 = 56.64 months; 53.17% of them were girls). Parents completed standardized assessments of parenting stress, children’s problem behaviors, and the risk of learning disorders. An autoregressive cross-lagged panel model within a half-longitudinal framework was employed to test the hypothesized mediation model. Results revealed that parenting stress had a direct effect on both children’s problem behaviors and their risk of learning disorders. Moreover, children’s problem behaviors partially mediated the longitudinal association between parenting stress and learning disorder risk. Specifically, the indirect effect size was statistically significant (*β* = 0.022, *p* = 0.025), indicating that increased parenting stress contributed to approximately a 2.2% increase in the risk of learning disorders through elevated problem behaviors. Theoretically, these findings underscore the critical role of child behavioral adjustment as a mechanism through which parenting stress may influence developmental risk. The results highlight the potential benefits of reducing parenting stress and addressing children’s behavioral difficulties to prevent early learning problems. However, the study relied exclusively on parent-reported data, which may introduce shared method variance and reporting bias; future research should incorporate multiple informants and objective behavioral assessments.

## 1. Introduction

Parenting stress in child-rearing refers to the pressure experienced by parents in fulfilling their roles and parenting tasks, influenced by personal characteristics, child attributes, and adverse parent–child interactions ([2]). According to the ecological systems theory, parenting stress is a significant familial factor within the micro-ecological system that closely relates to various behavioral problems in children ([89]). Problem behaviors are critical indicators of children’s social adaptation and typically encompass externalizing behaviors (e.g., aggression and delinquency) and internalizing behaviors (e.g., anxiety, depression, and social withdrawal) ([57]). Early problem behaviors in children can significantly negatively impact their peer relationships and subsequent academic performance, as well as school adjustment ([4]). Research indicates that problem behaviors among Chinese children have increased recently ([17]). High levels of parenting stress can lead to a range of negative emotional and behavioral responses from parents ([88]), which not only affects family functioning and quality of life but also exacerbates children’s emotional and behavioral issues ([24]).

The risk of learning disorders in early childhood refers to the presence of atypical early signs in preschool-aged children that deviate from normal development, such as delayed language acquisition, difficulties in counting, poor motor coordination, memory deficits, and hyperactivity ([87]). Children at risk of learning disorders often exhibit learning difficulties that do not match their intellectual developmental potential, with regular educational counseling ([38]; [87]). [9] ([9]) found that learning disorders can lead to significant and quantifiable underperformance relative to age-appropriate expectations, and without appropriate intervention measures, this discrepancy may adversely affect academic achievement and daily functioning. Meanwhile, [109] ([109]) found that the family environment and parenting styles significantly influence the occurrence and development of learning disorders in children.

Family Systems Theory posits that family systems are dynamic and subject to change at any level, where alterations at one level can provoke further changes in individuals, relationships, and the overall family system ([26]). The continuity of child development is embedded in the relationship between the child and the family system or caregivers, with parents being the most immediate contacts for preschool-aged children within this system ([105]). Parenting is regarded as one of the most critical factors in early childhood development, as it profoundly shapes the quality of parent–child interactions through daily caregiving behaviors—such as emotional responsiveness, behavioral regulation, and supportive communication ([11])—thereby influencing children’s social competence, emotion regulation, and cognitive developmental trajectories ([98]). However, although there has been extensive research on the impact of parenting stress on children’s development, there is still limited empirical research on the mediating mechanism between problem behaviors and the risk of learning disorders in a longitudinal framework. This lack of knowledge limits our understanding of the causal pathway and intervention nodes between the two.

However, current research often treats parenting stress as a static antecedent variable, primarily examining the immediate and delayed effects of baseline parenting stress on children’s problem behaviors and learning disorders while neglecting the possible dynamic temporal associations between them. This limitation constrains our in-depth understanding of how parenting stress influences developmental risks over time through child behavioral pathways. Therefore, identifying its underlying mechanisms is crucial for developing theoretically grounded intervention strategies to mitigate the adverse effects of parenting stress on child development. Moreover, cultural factors may further moderate the pathways through which parenting stress impacts child development. Within the Chinese cultural context, family structure, parenting beliefs, and access to educational resources differ significantly from those in Western societies. For instance, Chinese parents generally face higher educational expectations and greater pressure in child-rearing responsibilities ([18]). Thus, by focusing on Chinese preschool-aged children, this study can provide culturally representative empirical evidence on the association between parenting stress and child development, contributing to global research on this topic. The findings hold significant theoretical and practical implications for cross-cultural studies in child development.

## 2. Literature Review

### 2.1. Relationships Between Parenting Stress and Risk of Learning Disorder

Studies have shown that the prevalence of learning disorders in the general population ranges from 3% to 12% ([73]; [97]), and the average age at which a child is diagnosed with learning disorders is 9 years old (grades 3–4 of elementary school) ([106]). However, scholars widely recognize that learning disorders emerge at the kindergarten level and persist into adulthood ([91]; [108]). [10] ([10]) found that 5.7% of preschool children were at risk of developing specific learning disorders, while factors such as low fathers’ education were associated with an increased risk of developing learning disorders. Research suggests that if children at risk of learning disorders are identified in the preschool years and provided with appropriate intervention programs, the likelihood that they will be diagnosed with specific learning disorders during the school years is significantly reduced ([63]; [86]). The benefits of intervening in children’s early risk of learning disorders extend beyond the academic realm, with research finding that early diagnosis and intervention can help to ameliorate the learning disorders condition and its negative impact on children’s learning, socialization, and emotional development ([90]). In summary, exploring the factors that influence preschoolers’ risk of learning disorders may help intervene promptly and prevent later academic and social difficulties.

Parenting stress is a prevalent experience that every parent encounters following the birth of a child ([28]). Family Systems Theory posits that the family operates as an “emotional unit”, in which the emotions, behaviors, and thoughts of individuals are affected by other members within this interconnected relational network ([26]). Among these members, parents act as external factors that directly interact with children, and parenting practices are recognized as crucial determinants of child development ([105]). Research indicates that developmental needs demand more excellent care and parent–child interaction during early childhood, making parents more susceptible to stress ([19]). A study conducted on Chinese parents of preschool children revealed that 28% of Chinese parents were at high parenting stress levels ([18]). The longitudinal study by [43] ([43]) revealed that higher levels of parenting stress predispose caregivers to adopt controlling parenting strategies, such as excessive intervention and coercive guidance. These controlling behaviors may undermine children’s autonomy and learning motivation, constraining their active exploration and self-regulatory capacities during learning processes, thereby indirectly contributing to poorer academic performance ([102]; [103]). For children with manifestations of learning disorders, [49] ([49]) found that parenting stress was equally capable of affecting their levels of academic and social adjustment. Simultaneously, research indicates that parenting stress not only adversely affects caregivers’ mental health, manifesting as increased anxiety, depression, and emotional exhaustion, but also compromises the quality of parent–child interactions by impairing caregivers’ emotional sensitivity and behavioral consistency ([64]). The deterioration in parent–child relationship quality may subsequently lead to difficulties in children’s cognitive regulation and behavioral adaptation, thereby elevating their risk of developing learning disabilities ([60]). Furthermore, the COVID-19 pandemic exacerbated caregiving burdens and intensified role conflicts, resulting in heightened parenting stress levels reported globally ([29]; [55]). Although empirical evidence remains limited in China, recent findings from Germany indicate a sustained psychological burden on families beyond the acute phases of the COVID-19 pandemic. A longitudinal study by [13] ([13]) revealed that parenting stress increased significantly over time despite the easing of restrictions, while a cross-sectional comparison by [81] ([81]) showed that children and adolescents reported markedly lower health-related quality of life and higher mental health problems during the pandemic compared to pre-pandemic cohorts. This pattern underscores the issue’s enduring nature and cross-cultural pervasiveness, further highlighting the critical importance and urgency of the current investigation.

### 2.2. Problem Behavior as a Mediator

The General Theory of Psychological Stress supports the notion that an interactive relationship exists between parenting stress and children’s social adaptation ([71]). The Spillover Hypothesis also emphasizes that when individuals experience high psychological and physical stress levels, negative emotions can “spill over” into other contexts ([32]). Elevated parenting stress often transfers to parental behaviors and emotional expressions, impairing parents’ ability to engage in positive parenting practices ([72]) and affecting their capacity to provide emotional and behavioral support to their children ([96]). This can lead to parental responses characterized by alienation and avoidance and, at times, varying degrees of child maltreatment ([96]). In such high-stress environments, children may internalize the stress they experience, potentially resulting in issues such as anxiety ([99]). Poor family functioning is a significant contributor to children’s problem behaviors, with factors such as family cohesion, communication, and overall family atmosphere being robust predictors of the emergence of these behaviors ([14]). [89] ([89]) found that parenting stress may indirectly influence children’s behavioral issues through parenting behaviors. Meanwhile, studies indicate that parenting stress has a positive predictive effect on internalizing issues, such as anxiety, depression, and withdrawal, in children aged 3 to 6 years ([45]), as well as on externalizing problems like hyperactivity, aggression, and oppositional behaviors ([51]; [107]).

Regarding the association between children’s problem behaviors and risk of learning disorders, a large longitudinal study demonstrated that behavioral problems significantly predicted the expression of learning disorder symptoms from kindergarten through fifth grade ([46]). Furthermore, research has demonstrated significant differences in the associations between various types of learning disorders and internalizing problems such as anxiety, depression, and attention difficulties in children ([83]). Differences exist in the relationship between distinct learning disorder subtypes and externalizing behavior problems, such as hyperactivity and peer interactions ([74]). The comorbidity of learning disorders with other disorders, such as autism spectrum disorder or attention-deficit/hyperactivity disorder (ADHD), is notably high ([50]; [67]). Among these, hyperactivity emerges as a central behavioral constraint factor impacting children with learning disorders ([30]), characterized by difficulties in sustaining attention, high vulnerability to distractions, and frequent impulsive behaviors, which negatively affect their working memory and organizational skills ([21]), ultimately leading to heightened levels of risk for learning disorders. A review of the literature highlights that learning disabilities and problem behaviors frequently co-occur. However, the majority of studies focus exclusively on whether the relationship is reciprocal or unidirectional, with minimal attention provided to standard variables that may influence both ([36]). Additionally, most research tends to examine variables at the child level ([36]). Therefore, the present study hypothesized that parenting stress acts through preschoolers’ problem behaviors to contribute to their risk for learning disorders.

### 2.3. The Present Study

Previous studies have extensively explored the relationship between parenting stress, preschool children’s problem behavior, and the risk of learning disorders. However, the longitudinal mechanisms of action between these variables remain unclear. In particular, there is a lack of systematic empirical research on whether problem behavior mediates the influence of parenting stress on preschoolers’ risk of learning disorders. According to the Family Systems Theory ([26]) and Spillover Hypothesis Theory ([32]), parenting behaviors in the home environment influence children’s behavioral and cognitive development. That is, parenting stress may not only predict a young child’s risk for learning disorders ([3]) but may also further exacerbate the risk for learning disorders by influencing a young child’s level of problem behavior ([74]; [83]). Therefore, based on the existing literature, the present study used two waves of follow-up data to explore the temporal dynamic relationship between parenting stress, preschoolers’ problem behavior, and the risk of learning disorders through a half-longitudinal mediation model and to test the mediating role of problem behavior in it. In summary, the following core hypotheses are proposed: (1) parenting stress positively predicts preschoolers’ risk of learning disorders, and (2) preschoolers’ problem behavior mediates the relationship between parenting stress and the risk of learning disorders. Based on the above hypotheses, this study constructed a conceptual model for observing the relationship between variables (see Figure 1).

In reviewing the literature, family socioeconomic status (SES), age, and gender typically influence children’s problem behavior and risk of learning disorders. For example, the gene–socioeconomic status interaction hypothesis of cognition and learning proposes that family socioeconomic status influences children’s behavior and learning development ([39]; [93]). Specifically, this hypothesis suggests that genetic influences on cognitive and academic outcomes are more fully expressed in children from higher-SES families, whereas adverse environmental conditions in lower-SES families may constrain the realization of genetic potential ([94]; [95]). This perspective highlights the critical role of family context in moderating the interplay between biological predispositions and learning trajectories. A meta-analysis showed that SES affects children’s problem behavior and academic achievement ([76]). According to the graphical causality model, when a variable affects both hyperactive behavior and the risk of learning disorder, it should be considered a “backdoor path” that needs to be cut off ([82]). Thus, socioeconomic status may be a “backdoor path” that needs further control in the temporal relationship between parenting stress, preschoolers’ problem behavior, and the risk of learning disorders. A similar situation exists regarding the age and gender of children. [101] ([101]) found that boys may exhibit more disruptive, hyperactive, and impulsive behaviors than girls. In a clustered randomized controlled trial, [8] ([8]) found peer-reported effects of problem behavior, such as hyperactivity, to be significant only in a group of boys. [66] ([66]) assessed 1633 children in grades 3–4 using the DSM-5, and they found that boys and girls performed differently across learning disorders subtypes, with boys exhibiting higher rates of spelling deficits compared to girls, while girls were more likely to be impaired in arithmetic. A longitudinal study by [5] ([5]) indicated that improvement in children’s early learning disorders did not predict their behavioral or emotional states 8 years later, but that levels of behavioral problems such as hyperactivity increased significantly as children with learning disorders aged in the group. In conclusion, this study used family socioeconomic status (SES), age, and gender of young children as control variables to obtain more reliable results through relatively more rigorous data analysis.

## 3. Methods

### 3.1. Participants and Procedure

The study began with a sample of 313 children and their parents, ultimately including data from 284 children and their parents. The attrition rate is 9.27%. In the first wave (T1), the sample was n = 313; in the second (T2), n = 284. As can be seen, the sample number has not undergone significant alterations over time. We summarize the characteristics of the final sample, with parents’ ages ranging from 25 to 53 years (M = 34.95, SD = 4.12) and children’s ages ranging from 30 to 79 months (M = 56.64, SD = 10.70); additional characteristics are detailed in Table 1. The change in sample size was due to attrition caused by some of the children advancing to higher grades or transferring schools, with a 6-month time interval from T1 to T2. All data were collected within a single academic year, with the first wave conducted in October 2024 and the second wave in March 2025. This ensured consistency in contextual factors, including the relative stability of post-COVID-19 conditions in China during that period. To assess whether sample attrition introduced systematic bias, we conducted independent samples *t*-tests and chi-square tests comparing the retained participants (n = 284) with those lost to follow-up (n = 29) on key demographic and baseline variables, including parental age, child sex, and T1 parenting stress levels. The results revealed no statistically significant differences between groups on any of these variables (all *p*s > 0.05), indicating that the sample attrition did not substantially compromise the representativeness of the final sample. All data were collected through parent-completed questionnaires at home. Parents were instructed to complete the surveys independently and return them via digital submission, ensuring convenience and standardization. All participating children and parents were native Mandarin speakers, and the sample was drawn from kindergartens in Northeast China, a region identified as middle income based on the 2022 Northeast China Social Development Statistical Bulletin and national statistical criteria.

We established contact with parents through partnering kindergartens and obtained parental information with the assistance of the preschool administration. Before recruiting participants, parents were provided with a link to access an online informed consent form. This form outlined the rights of the participants, indicating that the related data would be used solely for research purposes and emphasizing that participation is voluntary. It explains that declining to participate or withdrawing from the study would not result in any negative consequences. After reviewing the informed consent, parents decided whether to agree to participate in the study. Questionnaires were administered only to participants who consented. The survey could only commence after obtaining parental consent. Families who met eligibility criteria but declined to participate were classified as non-participants, whereas those who withdrew after providing consent would be considered drop-outs. The language used for testing for all samples was Mandarin. The research team obtained appropriate ethical and procedural authorization to assess parents and children at different time points without conducting any interventions, ensuring the confidentiality of the data.

### 3.2. Instruments

#### 3.2.1. Parenting Stress

Due to the conceptual overlap between the “Difficult Child” dimension of the Parenting Stress Index-Short Form (PSI-SF) developed by [1] ([1]) and the child problem behaviors assessed in this study, this dimension was excluded to reduce potential multicollinearity, shared method variance, and measurement redundancy, which can bias parameter estimates and compromise construct validity in statistical models ([34]; [75]). Also, [44] ([44]) examined the psychometric properties of this questionnaire and concluded that it contains both parental distress and parent–child dysfunctional interaction factors. Therefore, the present study used two dimensions—parental distress and parent–child dysfunctional interaction—of the PSI-SF to measure subjects’ parenting stress with 24 items, rather than employing the full 36-item version of the PSI-SF. The scale utilizes a five-point Likert-type scoring system, ranging from 1 (strongly disagree) to 5 (strongly agree), with higher scores indicating greater levels of perceived parenting stress. The Chinese version of the PSI-SF has been utilized in research within Chinese populations, demonstrating good reliability and validity ([59]). To provide further context for interpreting elevated stress levels, parents whose total scores fell at or above the 90th percentile were classified as experiencing high parenting stress. In contrast, those below the 15th percentile were classified as low-stress respondents ([37]). The Cronbach’s α coefficients were 0.966 and 0.971 for the parental distress dimension and 0.965 and 0.972 for the parent–child dysfunctional interaction dimension in stages T1 and T2, respectively.

#### 3.2.2. Preschoolers’ Problem Behavior

The study utilized the parent-reported Strengths and Difficulties Questionnaire (SDQ), a 25-item measure designed for assessing children aged 4–10 years ([40]). This questionnaire included four difficulty dimensions: emotional problems, conduct problems, hyperactivity, and peer relationship problems (20 items), as well as one strengths dimension of prosocial behavior (5 items). For this study, only the four difficulty dimensions were used to measure children’s problematic behaviors, utilizing a three-point Likert-type scoring system ranging from 0 (not true) to 2 (certainly true), where higher scores indicated higher levels of problematic behavior. The Chinese version of the SDQ has shown good reliability and validity in research involving Chinese populations ([53]). Following standard cut-offs, scores of 5+ on emotional symptoms, 4+ on conduct and peer problems, and 7+ on hyperactivity were considered abnormal, aiding interpretation of severity within the sample ([53]). At T1 and T2, the Cronbach’s alpha coefficients for the emotional symptoms dimension were 0.749 and 0.827, respectively; for the conduct problems dimension, the coefficients were 0.745 and 0.792, respectively; for the hyperactivity dimension, they were 0.752 and 0.835, respectively. The Cronbach’s alpha coefficient for the peer problems dimension remained constant at 0.772 at both time points.

#### 3.2.3. Risk of Learning Disorder

The Preschool Learning Scale (PLSS), developed by [106] ([106]), was utilized in this study. The PLSS contains 38 items divided into seven dimensions: attention, memorization, visual perception, auditory perception, motor coordination, and mathematical concept. The primary caregiver assessed the questionnaire. It used a five-point Likert-type scoring system, ranging from 1 (never) to 5 (always), with higher scores indicating a greater risk of learning disorders in preschool children. Scores at or above the 90th percentile were used to identify children at high risk for learning disorders, enhancing the interpretation of elevated difficulty within the sample ([106]). At T1 and T2, Cronbach’s α coefficients for the attention dimension were 0.808 and 0.889, respectively; for the memorization dimension, the coefficients were 0.705 and 0.817, respectively; for the visual perception dimension, they were 0.823 and 0.857, respectively; for the auditory perception dimension, 0.749 and 0.846, respectively; for the motor coordination dimension, 0.797 and 0.831, respectively; for the verbal competence dimension, 0.900 and 0.922, respectively; and the mathematical concept dimension, 0.709 and 0.839, respectively.

### 3.3. Analytic Plan

Considering that only two time data points were collected in this study, it was impossible to construct a full longitudinal mediation model to estimate the model’s parameters ([56]). Therefore, this study established a half-longitudinal mediation model ([22]) with the Parenting Stress Index (PSI) mean score as the independent variable, the Strengths and Difficulties Questionnaire (SDQ) difficulties mean score as the mediator, and the Preschool Learning Skills Scale (PLSS) mean score as the dependent variable. The inferential capacity of semi-longitudinal designs was markedly superior to that of cross-sectional mediation analyses, as both the mediator and outcome variables were controlled for their prior levels, allowing for a more precise examination of the impacts resulting from changes in the mediator and outcome variables ([56]).

Researchers have argued that within the framework of classical measurement theory and structural equation modeling, explicit variable models used to compare changes in variables in the time dimension also assume intertemporal stability of the measurement structure ([56]; [62]). In addition, [79] ([79]), in their systematic review of intertemporal measurement research, also noted that even when scale mean scores are used, the interpretive validity of findings may still be compromised if there is a structural or comprehension bias of the measurement instrument at different points in time. Because all of the studies in this review measured more abstract mental constructs, longitudinal analyses using mean scale scores must assume equivalence. Therefore, the scales were first tested for measurement invariance to ensure that the model consisting of the explicit variables was based on a stable measurement structure. Before proceeding with measurement invariance testing, we first evaluated potential common method bias (CMB), which can arise when all data are collected using the same source. To address this, we conducted confirmatory factor analyses (CFAs) with a single latent method factor at both T1 and T2, following the recommendation of [75] ([75]). Second, the appropriateness of a measure model in which all variables at both time points are correlated with each other was tested. The stability of variables across time was examined through correlations. In addition, the distributional characteristics of all observed variables were assessed using skewness and kurtosis statistics to determine the degree of normality. To further ensure the robustness of the modeling results, we examined potential multicollinearity among the predictors at T1 by calculating variance inflation factors (VIF). Third, a semi-longitudinal analysis was used to estimate the mediation model (PS → PPB → RLD, Figure 1), which is our main focus. Based on static assumptions, the mediation test within the semi-longitudinal design relies on using SEM to create an alternative parameter that estimates the indirect effects of the predictor on the outcome through the mediator at the two time points. This parameter represents the product of the two effects: the effect of the predictor on the mediator (a) and the effect of the mediator on the outcome (b) over the period from T1 to T2. To assess whether the product ab (the indirect effect) is statistically significant, we conducted a bootstrap sampling analysis with 5000 simulations. According to this procedure, if the indirect effect’s 95% confidence interval (CI) does not include zero, then the effect is considered significant ([56]; [78]).

Several fit indices were used to assess model fit: *x*^2^ and its ratio with the degrees of freedom (*x*^2^/*df*), the Comparative Fit Index (CFI), the Tucker–Lewis Index (TLI), the Root Mean Square Error of Approximation (RMSEA), and the Standardized Root Mean Square Residual (SRMR). A good fit of the model to the data is also indicated when the *x*^2^/*df* (degrees of freedom) ratio is less than three ([52]). For CFI, values above 0.95 are preferred, and values close to 0.90 are considered acceptable ([6]; [52]). TLI values above 0.95 suggest a good fit, while values above 0.90 are acceptable ([7]). RMSEA values below 0.05 reveal a good fit, whereas values between 0.05 and 0.08 reveal an acceptable fit ([12]). SRMR values less than 0.08 generally reflect an acceptable model fit ([48]). The software packages used for data analysis and processing were IBM SPSS Statistics 26.0 and Mplus 8.0 ([70]).

## 4. Results

### 4.1. Test of Common Method Bias

To examine the potential influence of common method bias (CMB) stemming from the use of parent-reported measures, we conducted a series of single-factor confirmatory factor analyses (CFAs) separately for the Time 1 and Time 2 data. For Time 1, all 84 measurement items (covering PSI, SDQ, and PLSS) were loaded onto a single latent factor. The model exhibited poor fit: *χ*^2^ (3402) = 8067.245, *p* < 0.001, RMSEA = 0.098 (90% CI [0.096, 0.101]), CFI = 0.475, TLI = 0.462, and SRMR = 0.154. Similarly, the single-factor model using all 81 items from Time 2 also showed unsatisfactory fit: *χ*^2^ (3159) = 12,869.108, *p* < 0.001, RMSEA = 0.104 (90% CI [0.102, 0.106]), CFI = 0.536, TLI = 0.524, and SRMR = 0.143. These results suggest that a single latent method factor cannot adequately explain the variance across the observed variables at either time point, indicating that common method bias is unlikely to pose a significant threat to the validity of the findings.

### 4.2. Descriptive Statistics and Correlation Analyses

Table 2 presents the descriptive statistics of the three constructs at T1 and T2. On a five-point Likert scale, the mean score of parenting stress was 2.178 at T1 (SD = 0.625) and 2.248 (SD = 0.678) at T2, and for preschoolers at risk of learning disorder, it was 2.134 (SD = 0.710) at T1 and 1.987 (SD = 0.708) at T2. On a three-point Likert scale, the mean score of preschoolers’ problem behavior was 0.496 at T1 (SD = 0.325) and 0.580 (SD = 0.455) at T2. All factors at both time points showed Cronbach’s alpha values ranging from 0.925 to 0.985, reaching acceptable reliability.

Additionally, the distributional characteristics of the observed variables were examined through skewness and kurtosis statistics. All values were within acceptable thresholds (|skew| < 2, |kurtosis| < 7) ([100]), indicating no substantial deviation from normality and supporting the use of maximum likelihood estimation in subsequent analyses.

The inter-factor correlations of the factors at T1 and T2 are presented in Table 3. The factors were found to positively correlate with each other at both time points. Correlations between the T1 variables and their T2 counterparts were moderate (*r* ranging from 0.148 to 0.730), indicating that these three constructs could be considered stable over time.

### 4.3. Multicollinearity Diagnostics

To evaluate potential multicollinearity among the observed indicators, Variance Inflation Factor (VIF) values were calculated for all T1 predictors, including 24 items from the PSI, 20 items from the SDQ, and 38 items from the PLSS. No VIF values exceeded the critical threshold of 10, which is generally regarded as the upper limit indicating serious multicollinearity ([69]). However, several PSI items showed moderate collinearity, with VIFs slightly above five. Specifically, PSI11 (VIF = 5.70), PSI15 (VIF = 5.53), and PSI21 (VIF = 5.22) were the highest. Since these indicators were modeled as latent constructs in the SEM analysis, multicollinearity is unlikely to have significantly influenced the structural estimates.

### 4.4. Measurement Invariance

To assess the longitudinal stability of measurement structures, measurement invariance was examined for three constructs: Parenting Stress Index (PSI), Strengths and Difficulties Questionnaire (SDQ), and Preschool Learning Skills Scale (PLSS) across two time points (T1 and T2). Since all three instruments were modeled using total scores in the subsequent structural equation modeling, a single-factor model was adopted for each construct. This modeling choice aligns with recommendations for parsimony and interpretability in longitudinal designs that employ summed scores as manifest indicators ([15]; [79]).

Configural models were first tested to establish baseline equivalence. The PSI model demonstrated an acceptable fit (*χ*^2^ = 2039.072, df = 1074, *p* < 0.001, RMSEA = 0.056, CFI = 0.947, and TLI = 0.944), as did the SDQ (*χ*^2^ = 1982.205, df = 748, *p* < 0.001, RMSEA = 0.076, CFI = 0.899, and TLI = 0.895) and the PLSS model (*χ*^2^ = 6484.380, df = 2776, *p* < 0.001, RMSEA = 0.069, CFI = 0.881, and TLI = 0.878). These results indicate that the overall factor structures were stable over time.

Metric invariance was tested by constraining factor loadings to equal T1 and T2. Full metric invariance was supported for PSI (Δ*χ*^2^(23) = 27.824, *p* = 0.223, and ΔCFI = 0.005), suggesting stable item-factor relations across time. In contrast, full metric invariance was not supported for SDQ (Δ*χ*^2^(7) = 68.831, *p* < 0.001, and ΔCFI = 0.006) and PLSS (Δ*χ*^2^(37) = 160.713, *p* < 0.001, and ΔCFI = 0.001), though the change in CFI for both remained within acceptable limits (ΔCFI < 0.01). Accordingly, partial metric invariance was established for SDQ and PLSS by relaxing a few constraints based on modification indices. The final models showed acceptable fit: PSI (*χ*^2^ = 2646.951, df = 1098, RMSEA = 0.070, CFI = 0.915, and TLI = 0.913), SDQ (*χ*^2^ = 1982.205, df = 748, RMSEA = 0.076, CFI = 0.899, and TLI = 0.895), and PLSS (*χ*^2^ = 6556.487, df = 2811, RMSEA = 0.068, CFI = 0.880, and TLI = 0.878). These results support the establishment of a full metric invariance for PSI and a partial metric invariance for SDQ and PLSS across time. See Table 4 for details.

### 4.5. Mediation Analysis

Due to the inability to test a comprehensive mediation model, we first evaluate a semi-longitudinal mediation model examining the hypothesized indirect effect of parenting stress (PS) at T1 on the risk of learning disorders (RLD) at T2 via preschoolers’ problem behaviors (PPB). The results indicate that the hypothesized model, which posits a pathway from PS (T1) to RLD (T2), fits the data well: the *x*^2^/*df* ratio = 1.194, the CFI was 0.996, the TLI was 0.991, the RMSEA value = 0.026, and the SRMR value = 0.026. Thus, our two-wave semi-longitudinal mediation model effectively captures the mediating effect of PPB in the relationship between PS and RLD over time, as detailed in Table 5.

As shown in Figure 2, the hypothesis that PPB mediates the relationship between PS at T1 and RLD at T2 is confirmed. There is an indirect effect through preschoolers’ problem behavior: PS at T1 is positively related to PPB at T2 (a = 0.109, *p* < 0.01), and PPB at T1 is related to RLD (b = 0.203, *p* < 0.001). The bootstrapping results reveal that the mediator effect of PPB resulted in significant indirect relationships between PS at T1 and RLD at T2 (a*b = 0.022, *p* = 0.025; 95% CI: 0.003, 0.041).

## 5. Discussion

Guided by the Family System Theory ([26]) and the Family Stress Model ([23]), this study adopted a semi-longitudinal mediational model to systematically examine the cross-time influence mechanism of parenting stress on the risk of learning disorders through children’s problem behaviors. The results showed that parenting stress at Time Point T1 significantly predicted children’s problem behaviors at Time Point T2 (a = 0.109, *p* < 0.01), and problem behaviors further significantly predicted children’s risk of learning disorders at T2 (b = 0.203, *p* < 0.001), indicating that this mechanism has a moderate effect size ([35]). The Bootstrap analysis further supported the robustness of this indirect effect (a × b = 0.022, *p* = 0.025, 95% CI = [0.003, 0.041]), suggesting that children’s problem behaviors play a significant mediating role between the two. This finding verifies the potential influence of family stress on children’s cognitive adaptation risks from the perspective of temporal dynamics, emphasizing the viewpoint that situational family stress can affect children’s development outcomes through proximal parent-child interactions ([23]). It also provides new empirical support for the extension of the Spillover Hypothesis ([32]) in the research on children’s behavior-learning risks. Although the value of the indirect effect is relatively small ([35]), its statistical significance indicates that the observed effect is unlikely to be due to chance. However, due to the small effect size, caution is warranted when interpreting its practical or clinical relevance ([20]). Especially in the process of early identification and intervention for children on the verge of development, more attention should be paid to the linkage effect between family stress management and children’s problem behaviors. These findings suggest that high levels of parenting stress are associated with increased problem behaviors in preschoolers, which in turn contribute to a higher risk of developing learning disorders. These results raise several theoretical and educational practice issues that merit further discussion, and they highlight the need for future studies to incorporate more objective indicators of children’s behaviors and academic achievements through longitudinal follow-up research.

### 5.1. The Effect of Parenting Stress on Preschoolers’ Risk of Learning Disorder

According to the Family Stress Model, parenting stress is a natural part of the parenting experience, and it occurs when children’s needs exceed their parents’ expectations and actual resources ([23]), which can occur in multiple domains related to parenting ([27]). Numerous studies have concluded that parents of children with signs of learning disorders tend to face higher levels of parenting stress than parents of typically developing children ([47]; [65]; [68]), and may even be chronically impacted by excessive parenting stress ([42]), but not enough attention has been paid to whether parenting stress is a risk factor for signs of learning disorders in early childhood. However, due to the important influence of the family environment on child development ([26]) and the actual situation of high levels of parenting stress among Chinese parents today ([18]), the association between parenting stress and the risk of early learning disorders among preschoolers deserves more attention from researchers.

Our first hypothesis (H1), that parenting stress positively predicts the risk of learning disorders in preschoolers, has been validated. This study advances the understanding of preschool children’s learning development mechanisms by integrating family systems theory and family stress modeling. It examines the negative impact of parenting stress in preschoolers on young children’s performance at risk for learning disorders. This finding echoes the argument proposed by [23] ([23]) that proximal parenting processes have a key role in translating situational stress into developmental risk for children. The exploratory study conducted by [84] ([84]) used focus group interviews to explore the pressures parents experience in caring for their children with learning disorders and their impact on the children. Thus, our study can provide more accurate quantitative evidence of how changes in parenting stress may impact the risk of learning disorders in early childhood over time. Meanwhile, a cross-sectional study of children with learning disorders found that changes in the level of academic adjustment of children with learning disorders were significantly influenced by parental stress ([49]), underscoring the association between parental emotions and children’s psychological characteristics and behavioral performance. Our study is partially consistent with the findings of [64] ([64]), which showed that parental stress caused children to show more learning disorders. In summary, our study contributes novel insights into the existing body of research by highlighting the longitudinal mediating role of preschoolers’ problem behavior in the relationship between parenting stress and the risk of learning disorders.

### 5.2. The Mediating Role of Preschoolers’ Problem Behavior

The second hypothesis (H2), which posits that preschoolers’ problem behavior mediates the relationship between parenting stress and risk for learning disorders, has been validated, again supporting family systems theory ([26]). According to the results of this study, parenting stress can increase problem behavior in preschool children over time. It can eventually lead to more severe manifestations of risk for learning disorders in children. Although the current empirical research on this topic is still relatively limited, the existing research findings have preliminarily shown consistency, and the findings of our study are also in line with the conclusions of some scholars ([45]; [51]; [107]). They have similarly emphasized the association between parenting stress and preschoolers’ problem behavior, suggesting that increased parenting stress corresponds to higher levels of problem behavior in young children. Similarly, existing research examining the relationship between problem behavior and the manifestations of learning disorders in preschoolers echoes the findings of this study ([46]; [74]; [83]). Moreover, in terms of child development, our findings also remain generally consistent with previous studies that have revealed the negative role played by parenting stress on preschool children’s development ([14]; [51]; [99]).

From the perspective of the family system, this study further reveals how parenting stress indirectly affects the risk of learning disabilities in young children through their problem behaviors, supporting the view that individual development is influenced by the dynamic interactions within the family environment ([26]). Specifically, the study found that when parents are under high parenting stress, they may exhibit lower emotional sensitivity and consistency in their daily parenting, which in turn is detrimental to the development of children’s emotional regulation and behavioral adaptation, increasing their risk of learning difficulties. This result expands the understanding of the “stress–behavior–learning” pathway in existing research ([16]; [25]), emphasizing that intervention strategies should focus on the construction of the emotional regulation and support system within the family. According to the spillover hypothesis ([32]), excessive psychological stress experienced by individuals in one situation may lead to the transfer of negative emotions to other situations, thereby affecting their functional performance in different domains ([99]). Specifically, the stress and negative emotions accumulated by parents in the family may be transmitted to children through parent–child interactions, causing them to internalize these stresses and exhibit emotional and behavioral problems ([33]). These problem behaviors not only affect children’s social adaptation ([31]), but they may even develop into learning disabilities in severe cases ([54]; [104]). In view of this, intervention strategies should focus on the construction of the emotional regulation and support system within the family. Research shows that improving family functioning and enhancing emotional regulation abilities can help alleviate parents’ stress levels, thereby promoting children’s behavioral and learning development ([41]; [71]). Therefore, future interventions should focus on enhancing the family’s emotional regulation capabilities and establishing an effective support system to mitigate the negative impact of parenting stress on children’s development.

It is worth noting that although parenting stress is essentially a subjective experience and is appropriately measured through parental self-assessment, the problem behaviors of young children and the risk of learning disabilities in this study are also based on parental reports, which may introduce subjective biases or common method variance. Future research could attempt to incorporate objective methods such as teacher evaluations, third-party observations, or clinical screenings to improve the accuracy of the assessment of children’s behavioral characteristics and learning risks, thereby enhancing the external validity of the research conclusions and their practical operability. Longitudinal designs are particularly beneficial for clarifying predictive directions and the reciprocal relationships between constructs ([92]). It is evident that the results of the present study further validate the parental effects model of family systems theory, whereby parenting stress mediates through preschoolers’ problem behaviors, which in turn affect preschoolers’ learning disorders risk profiles. There are no comprehensive studies that combine these three variables, which emphasizes the novelty of our findings. The longitudinal design of this study facilitates a more in-depth exploration of the relationships studied and improves the reliability of the findings compared to the cross-sectional approach.

## 6. Limitations and Future Study Directions

Although the present study obtained some interesting findings regarding the longitudinal relationship between parenting stress, preschoolers’ problem behavior, and the risk of learning disorders, two limitations in its design and data analysis point the way for future research. First, scholars have recommended utilizing at least three-wave panel data to ascertain longitudinal mediation effects ([77]) accurately. However, only two waves of data are collected in our study, which only allows for the assessment of semi-longitudinal mediation. This approach complicates evaluating whether the mediation effects are partial or complete ([85]). Therefore, future studies are encouraged to collect at least three waves of data to verify the stability and directionality of the effects found in this study. Second, all data were based on parental self-reports. While valuable for capturing subjective experiences, such data raise the possibility of common method bias. Future research should consider including additional informants (e.g., teachers) or objective measures (e.g., standardized assessments) to reduce this risk and enhance the validity of findings. Third, although the indirect effect observed in this study was statistically significant, its small effect size (explaining only 2.2% of the variance) indicates limited practical relevance and warrants cautious interpretation ([20]). It is likely that other factors, such as parenting style ([80]), executive function in children ([58]), or socioeconomic context ([61]), play more substantial roles in explaining the relationship between parenting stress and children’s developmental risk. Future research should explore these potential moderators or mediators to build a more comprehensive model.

## 7. Conclusions

Parenting stress is a common risk factor affecting child development. However, the topic of whether parenting stress hurts problem behavior and the risk of learning disorders in preschoolers remains controversial. Our findings complement the literature on the longitudinal relationship between parenting stress, preschoolers’ problem behavior, and risk for learning disorders. At the same time, our study finds that parenting stress significantly predicted the level of problem behavior and risk of learning disorders in preschoolers and that problem behavior mediated the longitudinal relationship between parenting stress and risk of learning disorders. Future research should further explore the specific aspects of children’s development that are influenced by parenting stress through longitudinal and interactive mechanisms, such as emotional regulation, executive functions, and learning abilities. In particular, it is recommended to adopt more objective research strategies, including multiple information sources (such as parental and teacher reports) and a combination of multiple methods (such as behavioral observations and physiological indicators) to deeply reveal the pathways through which family stress affects children’s development. This will provide a more robust empirical basis for the scientific formulation of family education intervention programs. This could help parents better emphasize the impact of their parenting stress on children’s developmental problems and avoid placing children in adverse developmental environments.

## Figures and Tables

**Figure 1 behavsci-15-00785-f001:**
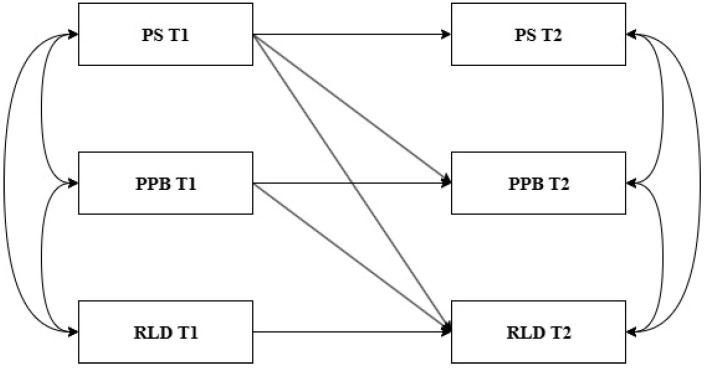
The hypothesized model was to be tested in the study. Abbreviation: Time 1 (T1), Time 2 (T2), parenting stress (PS), preschoolers’ problem behavior (PPB), risk of learning disorder (RLD); gender, monthly age of children, and family SES at T1 were included as covariates.

**Figure 2 behavsci-15-00785-f002:**
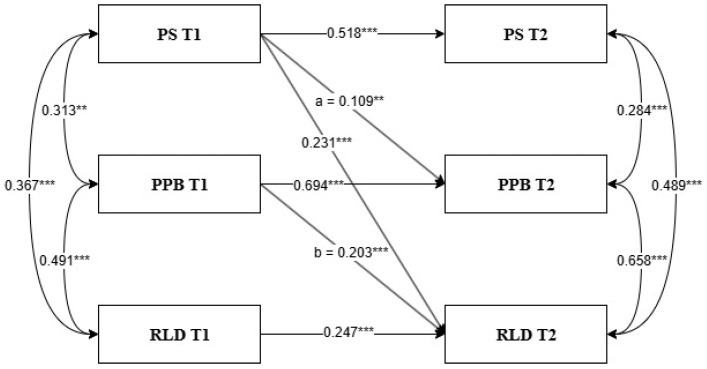
Half-longitudinal mediation model for testing the mediation role of PPB between PS at T1 and RLD at T2. Notes: values are standardized coefficients; ** *p* < 0.01, *** *p* < 0.001. Abbreviation: Time 1 (T1), Time 2 (T2), parenting stress (PS), preschoolers’ problem behavior (PPB), risk of learning disorder (RLD).

**Table 1 behavsci-15-00785-t001:** Background information on mothers and children (n = 284).

Demographics	Types	Frequency (%)
Sex of children	Boys	133 (46.83%)
	Girls	151 (53.17%)
Grades of children	Junior Class of Kindergarten	73 (25.70%)
	Middle Class of Kindergarten	125 (44.01%)
	Senior Class of Kindergarten	86 (30.28%)
One child in the family	Yes	208 (73.24%)
	No	76 (26.76%)
Parental education	Junior high school education and below	36 (12.68%)
	Senior high school education	37 (13.03%)
	Post-secondary education	84 (29.58%)
	Bachelor’s degree and above	127 (44.72%)
Parental working hours	Out of work	67 (23.59%)
	Less than 8 h	81 (28.52%)
	8~10 h	111 (39.08%)
	More than 10 h	25 (8.80%)
Family annual income	RMB 50,000 and below	37 (13.03%)
	RMB 60,000~100,000	78 (27.46%)
	RMB 110,000~150,000	80 (28.17%)
	RMB 160,000~200,000	33 (11.62%)
	RMB 200,000 and above	56 (19.72%)

**Table 2 behavsci-15-00785-t002:** Descriptive statistics and reliabilities at T1 and T2 (n = 284).

	T1				T2			
	M	SD	Range	Cronbach’s α	M	SD	Range	Cronbach’s α
PS	2.178	0.625	1–4.54	0.982	2.248	0.678	1–4.88	0.985
PPB	0.496	0.325	0–1.25	0.925	0.580	0.455	0–2.00	0.939
RLD	2.134	0.710	1–4.50	0.963	1.987	0.708	1–4.45	0.975

Abbreviation: Time 1 (T1), Time 2 (T2), parenting stress (PS), preschoolers’ problem behavior (PPB), and risk of learning disorder (RLD).

**Table 3 behavsci-15-00785-t003:** Inter-factor correlations (n = 284).

	1	2	3	4	5	6
1. Parenting stress_T1	-					
2. Parenting stress_T2	0.518 ***	-				
3. Preschoolers’ problem behavior_T1	0.292 ***	0.188 **	-			
4. Preschoolers’ problem behavior_T2	0.310 ***	0.351 ***	0.730 ***	-		
5. Risk of learning disorder_T1	0.333 ***	0.148 *	0.527 ***	0.360 ***	-	
6. Risk of learning disorder_T2	0.374 ***	0.553 ***	0.420 ***	0.704 ***	0.397 ***	-

Notes: T1: Time 1, T2: Time 2; * *p* < 0.05, ** *p* < 0.01, *** *p* < 0.001.

**Table 4 behavsci-15-00785-t004:** Tests of measurement invariance of the constructs across time.

Scale	Invariance	*x* ^2^	df	CFI	TLI	RMSEA	Comparison	Δ*x*^2^	*p*-Value	ΔCFI	ΔRMSEA
PS											
	(1) Configural	2419.426 *	1081	0.978	0.977	0.066					
	(2) Metric	2098.952 *	1104	0.983	0.983	0.056	(2) vs. (1)	27.824	0.223	0.005	−0.010
PPB											
	(4) Configural	1653.574 *	741	0.941	0.938	0.066					
	(5) Metric	1553.993 *	760	0.949	0.948	0.061	(5) vs. (4)	56.774	*p* < 0.001	0.008	−0.005
	(6) Partial metric	1510.373 *	753	0.951	0.950	0.060	(6) vs. (4)	14.676	0.260	0.002	0.001
RLD											
	(7) Configural	6484.380 *	2776	0.881	0.878	0.069					
	(8) Metric	6570.396 *	2813	0.880	0.878	0.069	(8) vs. (7)	160.713	*p* < 0.001	0.001	0.000
	(9) Partial metric	6556.487 *	2811	0.880	0.878	0.068	(9) vs. (7)	147.659	*p* < 0.001	0.000	−0.001

Note: * *p* < 0.05. Abbreviation: parenting stress (PS), preschoolers’ problem behavior (PPB), risk of learning disorder (RLD), degree of freedom (df), comparative fit index (CFI), tucker-lewis index (TLI), and root-mean-square error of approximation (RMSEA).

**Table 5 behavsci-15-00785-t005:** Path coefficients for the hypothesized mediation model.

Path	Mediation Model ^a^				
	*β*	SE	*p*-Value	*LLCI*	*ULCI*
Direct effect					
T1 PS → T2 PS	0.518	0.053	<0.001	0.414	0.618
T1 PS → T2 PPB	0.109	0.040	0.007	0.033	0.190
T1 PPB → T2 PPB	0.694	0.033	<0.001	0.624	0.755
T1 PS → T2 RLD	0.231	0.055	<0.001	0.118	0.333
T1 PPB → T2 RLD	0.203	0.051	<0.001	0.100	0.301
T1 RLD → T2 RLD	0.247	0.044	<0.001	0.164	0.335
Indirect effect					
PS → PPB → RLD	0.022	0.010	0.025	0.003	0.041

Note: *β* = standardized coefficients. Abbreviation: Time 1 (T1), Time 2 (T2), parenting stress (PS), preschoolers’ problem behavior (PPB), risk of learning disorder (RLD), standard error (SE), the lower limit of 95% confidence interval (*LLCI*), the upper limit of 95% confidence interval (*ULCI*), degree of freedom (*df*), comparative fit index (CFI), tucker-lewis index (TLI), and root-mean-square error of approximation (RMSEA), and standardized root mean square residual (SRMR). ^a^ The goodness for Model: *x*^2^ = 17.914, *df* = 15, *x*^2^/*df* = 1.194, CFI = 0.996, TLI = 0.991, RMSEA = 0.026, and SRMR = 0.026; sex, monthly age of children, and family SES at T1 were included as covariates.

## Data Availability

The datasets during and/or analyzed during the current study are available from the corresponding author upon reasonable request.

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
