# Peer review of "Unraveling the Longitudinal Relationships Among Parenting Stress, Preschoolers’ Problem Behavior, and Risk of Learning Disorder"

_behavsci, 2025, doi:10.3390/bs15060785_

Round 1
Reviewer 1 Report
Comments and Suggestions for Authors
Dear authors,
I recently had the opportunity to review your manuscript 'Unraveling the Longitudinal Relationships Among Parental Stress, Preschoolers' Problem Behavior, and Risk of Learning Disorder'. While this is an interesting topic and your methodological approach is sound, there are several major and minor points that need to be adressed before this manuscript meets the quality criteria for publication.
1. Your results would benefit from a deeper interpretation - at the moment they are being discussed in a rather superficial way, which makes it difficult to understand how they can be transferred into practice. My overall suggestion are for the discussion and conclusion parts: You need to go into more detail when explaining results instead of mostly just citing other literature (which is in addition often a repetetion from the introduction; please also add different relevant literature to the discussion). You also did not mention any of the effect sizes in your discussion, however, this is crucial to understand if the found results are of clinical importance.
2. I also feel that you need to generally go into more detail when describing other studies (for instance in the introduction section, see some examples under minor points below).
3. You are referring to and measuring parenting stress. Parenting stress and parental stress are not neccessarily the same constructs. Parenting stress is stress that derives from being a parent, i.e., when parental resources and child care demands are imbalanced. Parental stress can be any form of stress happening to a parent. Also, when citing Abidin, parenting stress is the correct term and the PSI is the Parenting Stress Index and not Parental Stress Index. Please change the wording accordingly and I would strongly advice not to use the terms interchangeably but only speak of parenting stress. In this case, however, you need to make sure that the reference studies cited by you also refer to parenting stress as understood by Abidin.
Minor points refer to wordings/ language/ need for further explanation:
l. 55: "abnormal activity exhibited by children before school age" => what is meant by abnormal activity? Please explain.
l. 68: "thus, parenting is considered one of the most critical factors in child development" => please add a short statement why this is the case (impact on parent-child-interaction via parenting behaviors...)
l. 106: "Rogers et al. (2009) found that high parental stress caused parents to control their children's academics more, leading to poor academic performance." => needs to be better explained as this is not a self-explaining result. What is the pathway to explain this result?
l.109 ff. "In the meantime, research has shown that parenting stress affects the quality of parent-child interactions as well as children's manifestations of learning disorders while negatively impacting parents' mental health (Mohammadipour et al., 2021)." => please explain these pathways better. You could also add a statement referring to the COVID-19 pandemic, which has had an immense impact on parenting stress. I am not aware of studies from China surveying parenting stress during and after the pandemic, but investigations from other countries (e.g., Germany) have found an increase in parenting stress during and even after COVID-19. Maybe this could even strengthen your research question.
l.137 f.: incomplete sentence
l. 196 f. "with more boys than girls showing spelling deficits and more girls impaired in arithmetic". => should say being impaired
l.197 f.: "Liu et al. (2004) found that children with signs of malnutrition at age 3 were more aggressive or hyperactive at age eight and had more externalizing problems at age eleven." => not quite sure how children with signs of malnutrition align with yout study (relevance of cited study in this context)?
Methods: Recruitment: How exactly were the families recruited? Please mention how families were contacted, how did you get their information, how did you get in touch?
l. 228: "For those who decline to participate, the survey is terminated" => No. The survey can only start if parents consent to participate. The eligible families not consenting to participate are non-participants, otherwise they would be drop-outs. Please be more clear in your wording at this point, so the reader can undertand what you mean when you speak of your attrition rate.
ll. 235 ff.: The meaning of the sentence is not clear.
- SDQ: please mention the version (age specific version) you used.
- Analytic plan. Please be more precise and use variable names instead of construct names (e.g., PSI mean scores instead of parenting stress) - which were independent, dependent which was the mediating variable? This needs to be so clear that the model calculations could be replicated if someone had access to your dataset.
l. 377 ff. no need to explain that e.g., 0.996 is above 0.95 etc...
Discussion:
ll. 412 ff.: "These findings suggest that more significant parental stress leads to higher levels of problem behavior in preschoolers, which in turn leads to more risk conditions for learning disorders" => wording: what does more significant parental stress mean? I suppose you are referring to high or critically high parenting stress. Please only use the term significant when referencing a statistical characteristic.
l. 418: "and it occurs when parents' needs exceed their expectations and actual resources (Conger & Donnellan, 2007) (...)". => I am guessing that you mean children's needs, not parents' needs?
l. 447: "In summary, our study is a valuable addition to the results of existing studies" I would advice to rewrite this sentence - did you mean that your study's results are in line with most of the other existing studies/ adds valuable knowledge?
ll. 455 ff.: "Although similar studies are scarce in the literature, an important conclusion that can be drawn is that our findings are consistent with those of many scholars
(He et al., 2023; Kang et al., 2022; Zhang et al., 2023)" => seems contradictory, please rephrase
ll. 466 ff.: repetition of introduction, findings should be interpreted on a deeper level
Study limitations: explain which constructs should better be measured by a more objective apporach than parental report- supposedly not parenting stress (needs to be evaluated subjectively...)?
Conclusion: "Further research on the effects of parental stress on children's development and the underlying mechanisms would be a meaningful research
perspective and could bring important guiding value to family education". => too vague, which research precisely?
The language quality differs drastically between the manuscripts individual sections. I would advice to generally do a thorough language check on the whole manuscript. However, one specific: please do not change between tenses. The adequate tense for presenting original research is past tense.
Author Response
Point 1: Your results would benefit from a deeper interpretation - at the moment they are being discussed in a rather superficial way, which makes it difficult to understand how they can be transferred into practice. My overall suggestion are for the discussion and conclusion parts: You need to go into more detail when explaining results instead of mostly just citing other literature (which is in addition often a repetetion from the introduction; please also add different relevant literature to the discussion). You also did not mention any of the effect sizes in your discussion, however, this is crucial to understand if the found results are of clinical importance.
Response 1: Dear Reviewer, first of all, we would like to express our sincere gratitude for your valuable comments on our manuscript. Regarding the issues you raised, firstly, we have added the effect size of this study in the specific content of the discussion section. Secondly, we have made corresponding adjustments to the interpretation of the results and the elaboration of relevant arguments in the discussion. For the specific content, please refer to the highlighted part of the Discussion and Conclusions. Thank you again for your valuable feedback.
Point 2: I also feel that you need to generally go into more detail when describing other studies (for instance in the introduction section, see some examples under minor points below).
Response 2: Dear Reviewer, first and foremost, we would like to extend our heartfelt gratitude for your valuable comments on our manuscript. We have carefully read and reflected on your viewpoints. Regarding each of the issues you mentioned, we have made adjustments in the corresponding sections. For detailed information, please refer to the responses for each individual point below.
Point 2.1: 55: "abnormal activity exhibited by children before school age" => what is meant by abnormal activity? Please explain.
Response 2.1: Dear Reviewer, first of all, we would like to express our sincere gratitude for your valuable comments on our manuscript. We have carefully read and considered your viewpoints. Regarding the issues you raised, first of all, we have confirmed the specific content of the original literature and found that our previous expression was inappropriate. The correct content should be "hyperactivity", which is one of the early signs of deviation from the normal behavior that children show in the preschool stage, instead of “abnormal activity”. Secondly, we apologize for the misunderstanding caused by our text expression. And we have made corresponding adjustments to the content in this regard. Please refer to the highlighted part (Lines 57 to 60). Thank you again for your valuable feedback.
Point 2.2: 68: "thus, parenting is considered one of the most critical factors in child development" => please add a short statement why this is the case (impact on parent-child-interaction via parenting behaviors...)
Response 2.2: Dear Reviewer, first of all, we are extremely grateful for the valuable comments you have made on our manuscript. We have carefully read and reflected on your viewpoints. In response to the issues you mentioned, we have adjusted the text in that part and added explanatory support for this viewpoint. Please refer to the highlighted part (Lines 71 to 79). Thank you again for your precious comments.
Point 2.3: 106: "Rogers et al. (2009) found that high parental stress caused parents to control their children's academics more, leading to poor academic performance." => needs to be better explained as this is not a self-explaining result. What is the pathway to explain this result?
Response 2.3: Dear Reviewer, first and foremost, we sincerely appreciate the valuable comments you provided on our manuscript. We have carefully read and given thought to your perspectives. Regarding the issues you raised, we have adjusted the text at that particular part and added explanatory support for this viewpoint. Please refer to the highlighted section (Lines 131 to 136). Thank you once again for your invaluable comments.
Point 2.4: 109 ff. "In the meantime, research has shown that parenting stress affects the quality of parent-child interactions as well as children's manifestations of learning disorders while negatively impacting parents' mental health (Mohammadipour et al., 2021)." => please explain these pathways better. You could also add a statement referring to the COVID-19 pandemic, which has had an immense impact on parenting stress. I am not aware of studies from China surveying parenting stress during and after the pandemic, but investigations from other countries (e.g., Germany) have found an increase in parenting stress during and even after COVID-19. Maybe this could even strengthen your research question.
Response 2.4: Dear Reviewer, first of all, we are truly grateful for the precious comments you have made on our manuscript. We have carefully read and reflected on your viewpoints. In response to the issues you mentioned, we have adjusted the text in this place and added explanatory support for the relevant arguments. Please refer to the highlighted part (Lines 138 to 152). Thank you again for your valuable comments.
Point 2.5: 137 f.: incomplete sentence
Response 2.5: Dear Reviewer, first and foremost, we are extremely grateful for the valuable comments you provided on our manuscript. In response to your concerns, after re-reading the content in that part, we have adjusted the expression of the transitional sentence. Please refer to the highlighted part (Lines 174 to 177) for the specific adjustments. Thank you again for your precious feedback.
Point 2.6: 196 f. "with more boys than girls showing spelling deficits and more girls impaired in arithmetic". => should say being impaired
Response 2.6: Dear Reviewer, first of all, we are sincerely grateful for the valuable comments you made on our manuscript. Regarding your questions, after re-reading the content of this part, we have adjusted the wording of the text here. Please refer to the highlighted part (Lines 239 to 240) for the specific adjustments. Thank you again for your precious feedback.
Point 2.7: 197 f.: "Liu et al. (2004) found that children with signs of malnutrition at age 3 were more aggressive or hyperactive at age eight and had more externalizing problems at age eleven." => not quite sure how children with signs of malnutrition align with yout study (relevance of cited study in this context)?
Response 2.7: Dear Reviewer, first of all, we are truly grateful for the valuable comments you offered on our manuscript. In response to your query, after re-reading the content of this part, we have deleted the citation in this place. Thank you again for your precious feedback.
Point 2.8: Methods: Recruitment: How exactly were the families recruited? Please mention how families were contacted, how did you get their information, how did you get in touch?
Response 2.8: Dear Reviewer, first and foremost, we sincerely appreciate the valuable comments you have made on our manuscript. In response to your question, we have adjusted the text in the section regarding the recruitment of research subjects. Please refer to the highlighted part (Lines 276 to 277) for the specific adjustments. Thank you again for your precious feedback.
Point 2.9: 228: "For those who decline to participate, the survey is terminated" => No. The survey can only start if parents consent to participate. The eligible families not consenting to participate are non-participants, otherwise they would be drop-outs. Please be more clear in your wording at this point, so the reader can undertand what you mean when you speak of your attrition rate.
Response 2.9: Dear Reviewer, first of all, we are extremely grateful for the valuable suggestions you have put forward regarding our manuscript. In response to the issue you raised, after rereading the relevant content, we have adjusted the wording of the text in that part. Please refer to the highlighted section (Lines 284 to 287) for the specific modifications. Thank you once again for your invaluable feedback.
Point 2.10: - SDQ: please mention the version (age specific version) you used.
Response 2.10: Dear Reviewer, first of all, we sincerely appreciate the valuable comments you made on our manuscript. Regarding the question you raised, we have adjusted the expression of the text in this part. Please refer to the highlighted content (Lines 312 to 314) for the specific adjustments. Thank you again for your precious feedback.
Point 2.11: Analytic plan. Please be more precise and use variable names instead of construct names (e.g., PSI mean scores instead of parenting stress) - which were independent, dependent which was the mediating variable? This needs to be so clear that the model calculations could be replicated if someone had access to your dataset.
Response 2.11: Dear Reviewer, first and foremost, we are truly grateful for the valuable comments you have provided on our manuscript. In response to your question, we have adjusted the wording of the text in this place. Please refer to the highlighted part (Lines 347 to 351) for the specific adjustments.Thank you again for your precious feedback.
Point 2.12: 377 ff. no need to explain that e.g., 0.996 is above 0.95 etc...
Response 2.12: Dear Reviewer, first and foremost, we are truly grateful for the valuable comments you have provided on our manuscript. In response to your question, we have adjusted the wording of the text in this place. Please refer to the highlighted part (Lines 481 to 482) for the specific adjustments.Thank you again for your precious feedback.
Point 2.13: 412 ff.: "These findings suggest that more significant parental stress leads to higher levels of problem behavior in preschoolers, which in turn leads to more risk conditions for learning disorders" => wording: what does more significant parental stress mean? I suppose you are referring to high or critically high parenting stress. Please only use the term significant when referencing a statistical characteristic.
Response 2.13: Dear Reviewer, first and foremost, we are truly grateful for the valuable comments you have provided on our manuscript. In response to your question, we have adjusted the wording of the text in this place. Please refer to the highlighted part (Lines 525 to 527) for the specific adjustments. Thank you again for your precious feedback.
Point 2.14: 418: "and it occurs when parents' needs exceed their expectations and actual resources (Conger & Donnellan, 2007) (...)". => I am guessing that you mean children's needs, not parents' needs?
Response 2.14: Dear Reviewer, first and foremost, we are truly grateful for the valuable comments you have provided on our manuscript. In response to your question, we have adjusted the wording of the text in this place. Please refer to the highlighted part (Lines 532 to 533) for the specific adjustments. Thank you again for your precious feedback.
Point 2.15: 447: "In summary, our study is a valuable addition to the results of existing studies" I would advice to rewrite this sentence - did you mean that your study's results are in line with most of the other existing studies/ adds valuable knowledge?
Response 2.15: Dear Reviewer, first and foremost, we are truly grateful for the valuable comments you have provided on our manuscript. In response to your question, we have adjusted the wording of the text in this place. Please refer to the highlighted part (Lines 561 to 564) for the specific adjustments. Thank you again for your precious feedback.
Point 2.16: 455 ff.: "Although similar studies are scarce in the literature, an important conclusion that can be drawn is that our findings are consistent with those of many scholars (He et al., 2023; Kang et al., 2022; Zhang et al., 2023)" => seems contradictory, please rephrase
Response 2.16: Dear Reviewer, first and foremost, we are truly grateful for the valuable comments you have provided on our manuscript. In response to your question, we have adjusted the wording of the text in this place. Please refer to the highlighted part (Lines 571 to 574) for the specific adjustments. Thank you again for your precious feedback.
Point 2.17: 466 ff.: repetition of introduction, findings should be interpreted on a deeper level Study limitations: explain which constructs should better be measured by a more objective apporach than parental report- supposedly not parenting stress (needs to be evaluated subjectively...)?
Response 2.17: Dear Reviewer, first and foremost, we are truly grateful for the valuable comments you have provided on our manuscript. In response to your question, we have adjusted the wording of the text in this place. Please refer to the highlighted part (Lines 583 to 616) for the specific adjustments. Thank you again for your precious feedback.
Point 2.18: Conclusion: "Further research on the effects of parental stress on children's development and the underlying mechanisms would be a meaningful research perspective and could bring important guiding value to family education". => too vague, which research precisely?
Response 2.18: Dear Reviewer, first and foremost, we are truly grateful for the valuable comments you have provided on our manuscript. In response to your question, we have adjusted the wording of the text in this place. Please refer to the highlighted part (Lines 650 to 658) for the specific adjustments. Thank you again for your precious feedback.
Point 3: You are referring to and measuring parenting stress. Parenting stress and parental stress are not neccessarily the same constructs. Parenting stress is stress that derives from being a parent, i.e., when parental resources and child care demands are imbalanced. Parental stress can be any form of stress happening to a parent. Also, when citing Abidin, parenting stress is the correct term and the PSI is the Parenting Stress Index and not Parental Stress Index. Please change the wording accordingly and I would strongly advice not to use the terms interchangeably but only speak of parenting stress. In this case, however, you need to make sure that the reference studies cited by you also refer to parenting stress as understood by Abidin.
Response 3: Dear Reviewer, first of all, we are extremely grateful for the valuable comments you have made on our manuscript. In response to your suggestion, we have re-examined the usage of this term throughout the text to ensure its correct application. Thank you again for your precious feedback.
Point 4: The language quality differs drastically between the manuscripts individual sections. I would advice to generally do a thorough language check on the whole manuscript. However, one specific: please do not change between tenses. The adequate tense for presenting original research is past tense.
Response 4: Dear Reviewer, first and foremost, we sincerely appreciate the valuable comments you've offered on our manuscript. In response to your suggestions, we have rechecked and sorted out the language of the text to ensure the correctness of the vocabulary, grammar, and tenses. Thank you once again for your precious feedback.
Reviewer 2 Report
Comments and Suggestions for Authors
Dear Editor,
I am writing to provide a review of the manuscript titled "Unraveling the Longitudinal Relationships Among Parental Stress, Preschoolers’ Problem Behavior, and Risk of Learning Disorder" submitted to Behavioral Sciences (behavsci-3620499).
Abstract: The abstract effectively summarizes the study's aims, methodology, key findings, and conclusions, offering a clear overview of the research. To enhance clarity, consider quantifying the direct and indirect effects identified, such as stating that "Parental stress increased the risk of learning disorders by X%." Additionally, briefly mention the study's limitations, such as the reliance on parental self-reports, to provide a balanced perspective.
Introduction: The introduction successfully establishes the significance of parental stress, problem behaviors, and learning disorders in early childhood development. It integrates parenting stress within an ecological systems theory framework, enhancing the study's context. To strengthen this section, clearly articulate gaps in existing literature by specifying the lack of longitudinal studies examining mediating mechanisms. Additionally, elaborate on the relevance and uniqueness of studying Chinese preschool children, considering cultural influences on parental stress or access to interventions.
Methods: The methods section provides detailed participant information, including age range and gender distribution, enhancing study transparency. It includes appropriate measures for parental stress, problem behavior, and risk of learning disorders, with clear justifications for their selection. Address the potential impact of the 9.27% attrition rate by discussing any notable differences between participants who remained and those who dropped out. Clarify the criteria for determining "middle-income level" in Northeast China, specifying specific income ranges or benchmarks. Lastly, include a brief description of the ethical considerations and approvals obtained for the study.
Results: The results section presents descriptive statistics and correlation analyses clearly, using appropriate statistical techniques such as semi-longitudinal mediation models to test the hypotheses. Provide more context for the descriptive statistics by discussing whether the observed means for parental stress, problem behavior, and risk of learning disorders are high, low, or typical compared to other studies or norms. Include a table showing standardized regression coefficients for direct and indirect effects in the mediation model using the Bootstrapping method, and address potential multicollinearity issues among predictor variables. Additionally, report Skewness and Kurtosis of the main variables and estimations of common method bias.
Discussion: The discussion effectively connects findings back to the Family Systems Theory and Spillover Hypothesis, reinforcing the theoretical framework. It acknowledges study limitations, such as reliance on parental self-reports, providing a balanced perspective. Expand on the implications for intervention and prevention strategies, suggesting effective intervention types to reduce parental stress and problem behaviors in preschoolers. Discuss the generalizability of findings to other cultural contexts or populations and suggest directions for future research, like incorporating teacher assessments or observational data to mitigate common method bias.
Conclusion: The conclusion succinctly summarizes the main findings, highlighting the study's significance. It reiterates the importance of addressing parental stress and problem behavior in preschoolers to prevent learning disorders. To strengthen this section, emphasize the potential impact of findings on family education and policy, offering practical recommendations for parents and educators. Conclude with a call to action, encouraging further investigation into the complex relationships among parental stress, child behavior, and learning outcomes.
References: Consider including more recent and relevant studies to support arguments and provide context.
Overall: The paper is well-written and addresses an important topic. It contributes valuable insights into the complex relationships between parental stress, problem behavior, and the risk of learning disorders in preschoolers. While using a semi-longitudinal mediation model is appropriate, acknowledging the limitations of this approach is essential, and future research with more time points is suggested. The study could benefit from incorporating more objective measures of child behavior and academic achievement to reduce common method bias.
I hope these comments and suggestions are helpful to the authors in refining their paper. Please feel free to contact me for further clarification on any points.
Thank you for the opportunity to review this manuscript.
Sincerely,
Author Response
Point 1: Abstract: The abstract effectively summarizes the study's aims, methodology, key findings, and conclusions, offering a clear overview of the research. To enhance clarity, consider quantifying the direct and indirect effects identified, such as stating that "Parental stress increased the risk of learning disorders by X%." Additionally, briefly mention the study's limitations, such as the reliance on parental self-reports, to provide a balanced perspective.
Response 1: Dear Reviewer, first of all, we are extremely grateful for the valuable comments you have made on our manuscript. We have carefully read and considered your viewpoints. In response to the issues you mentioned, we have adjusted the text of the abstract, as shown in the highlighted part (Lines 28 to 31, 34 to 36). Thank you again for your precious feedback.
Point 2: Introduction: The introduction successfully establishes the significance of parental stress, problem behaviors, and learning disorders in early childhood development. It integrates parenting stress within an ecological systems theory framework, enhancing the study's context. To strengthen this section, clearly articulate gaps in existing literature by specifying the lack of longitudinal studies examining mediating mechanisms. Additionally, elaborate on the relevance and uniqueness of studying Chinese preschool children, considering cultural influences on parental stress or access to interventions.
Response 2: Dear Reviewer, first of all, we are extremely grateful for the valuable comments you have made on our manuscript. We have carefully read and considered your viewpoints. In response to the issues you mentioned, we have adjusted the text of the introduction, as shown in the highlighted part (Lines 79 to 83, 87 to 100). Thank you again for your precious feedback.
Point 3: Methods: The methods section provides detailed participant information, including age range and gender distribution, enhancing study transparency. It includes appropriate measures for parental stress, problem behavior, and risk of learning disorders, with clear justifications for their selection. Address the potential impact of the 9.27% attrition rate by discussing any notable differences between participants who remained and those who dropped out. Clarify the criteria for determining "middle-income level" in Northeast China, specifying specific income ranges or benchmarks. Lastly, include a brief description of the ethical considerations and approvals obtained for the study.
Response 3: Dear Reviewer, first of all, we are extremely grateful for the valuable comments you have made on our manuscript. We have carefully read and considered your viewpoints. In response to the issues you mentioned, we have adjusted the text of the methods, as shown in the highlighted part (Lines 263 to 269, 273 to 275). Thank you again for your precious feedback
Point 4: Results: The results section presents descriptive statistics and correlation analyses clearly, using appropriate statistical techniques such as semi-longitudinal mediation models to test the hypotheses. Provide more context for the descriptive statistics by discussing whether the observed means for parental stress, problem behavior, and risk of learning disorders are high, low, or typical compared to other studies or norms. Include a table showing standardized regression coefficients for direct and indirect effects in the mediation model using the Bootstrapping method, and address potential multicollinearity issues among predictor variables. Additionally, report Skewness and Kurtosis of the main variables and estimations of common method bias.
Response 4: Dear Reviewer, first of all, we are extremely grateful for the valuable comments you have made on our manuscript. We have carefully read and considered your viewpoints. In response to the issues you mentioned, we have adjusted the text of the methods and results, as shown in the highlighted part (Lines 365 to 369, 372 to 376, 400 to 411, 420 to 424, 435 to 444). Thank you again for your precious feedback.
Point 5: Discussion: The discussion effectively connects findings back to the Family Systems Theory and Spillover Hypothesis, reinforcing the theoretical framework. It acknowledges study limitations, such as reliance on parental self-reports, providing a balanced perspective. Expand on the implications for intervention and prevention strategies, suggesting effective intervention types to reduce parental stress and problem behaviors in preschoolers. Discuss the generalizability of findings to other cultural contexts or populations and suggest directions for future research, like incorporating teacher assessments or observational data to mitigate common method bias.
Response 5: Dear Reviewer, first of all, we are extremely grateful for the valuable comments you have made on our manuscript. We have carefully read and considered your viewpoints. In response to the issues you mentioned, we have adjusted the text of the discussion, as shown in the highlighted part (Lines 505 to 525, 609 to 616). Thank you again for your precious feedback.
Point 6: Conclusion: The conclusion succinctly summarizes the main findings, highlighting the study's significance. It reiterates the importance of addressing parental stress and problem behavior in preschoolers to prevent learning disorders. To strengthen this section, emphasize the potential impact of findings on family education and policy, offering practical recommendations for parents and educators. Conclude with a call to action, encouraging further investigation into the complex relationships among parental stress, child behavior, and learning outcomes.
Response 6: Dear Reviewer, first of all, we are extremely grateful for the valuable comments you have made on our manuscript. We have carefully read and considered your viewpoints. In response to the issues you mentioned, we have adjusted the text of the conclusion, as shown in the highlighted part (Lines 650 to 658). Thank you again for your precious feedback.
Point 7: References: Consider including more recent and relevant studies to support arguments and provide context.
Response 7: Dear Reviewer, first and foremost, we are sincerely grateful for the valuable comments you have provided on our manuscript. We have carefully read and pondered over your perspectives. In response to your suggestions, we have incorporated more relevant and recent literature to support the arguments of our study. Thank you once again for your precious feedback.
Point 8: Overall: The paper is well-written and addresses an important topic. It contributes valuable insights into the complex relationships between parental stress, problem behavior, and the risk of learning disorders in preschoolers. While using a semi-longitudinal mediation model is appropriate, acknowledging the limitations of this approach is essential, and future research with more time points is suggested. The study could benefit from incorporating more objective measures of child behavior and academic achievement to reduce common method bias.
Response 8: Dear Reviewer, first of all, we are extremely grateful for the valuable comments you have made on our manuscript. In response to your suggestions, we will continue to conduct follow-up research in the future and attempt to incorporate more objective indicators of children's behaviors and academic achievements. Thank you again for your precious feedback.
Reviewer 3 Report
Comments and Suggestions for Authors
The present study advances the understanding of preschool children's learning development mechanisms by integrating family systems theory and family stress modeling. It examines the negative impact of parental stress in preschoolers on young children's performance at risk for learning disorders using a longitudinal design to explore the relationship between parental stress, preschoolers' problem behavior, and the risk of learning disorders It is well written and coherent with many methodological strengths.
Overall the article is well grounded and the literature review logically builds a case for the current study. The methods may be strengthened with the addition of the following two pieces of information:
- The description of procedures does not specify whether children were tested in a lab setting, in their homes, or within the school setting.
- Can the authors also please clarify whether all children were tested within the same calendar year, or was this sample built up over multiple years of testing? With the broader context of the global COVID-19 pandemic, this kind of temporal information seems relevant for the reader to contextualize the findings.
In addition, as I reviewed the discussion, many of the phrases used suggested extreme cases of parental stress, problem behaviors or learning disorders. However, looking at the Tables provided, there is limited information about whether this sample adequately captured a population that was representative of these dire outcomes. Either additional information in the Instruments section about any relevant cut-off values for “severe” levels of each variable, or a figure near Table 2 that transparently displays the spread of values for each variable would add important context to the findings. While the Mean, SD, and ranges in Table 2 are very well presented and helpful, they do not tell the full story of who is included in this sample.
Specific comments:
line 106-107 “Rogers et al. (2009) found that high parental stress caused parents to control their children's academics more, leading to poor academic performance.” – The words “caused” and “leading to” in this sentence are inappropriate as the study cited is cross sectional which means it cannot inspect causation or longitudinal changes in academics. Cross-sectional studies can only describe associations between variables. I also believe it is important to note in the text that the sample in that particular study was specific to children with ADHD.
Lines 182-184 “the gene-socioeconomic status interaction hypothesis of cognition and learning proposes that family socioeconomic status influences children's behavior and learning development (González et al., 2024; Tucker-Drob, 2009).” This description of the gene-socioeconomic status interaction hypothesis is leaving out a lot of critical context. If adding additional context about the gene-socioeconomic status interaction hypothesis is not a priority, the subsequent sentences in that paragraph are much stronger and may be sufficient.
Lines 236-238 “Considering the intersection of the difficult child dimension of the Parental Stress Index-Short Form (PSI-SF) developed by Abidin (1990) with the problem behavior subsequently measured in this study.” – This sentence does not make sense and should be revised.
Author Response
Point 1: The description of procedures does not specify whether children were tested in a lab setting, in their homes, or within the school setting.
Response 1: Dear Reviewer, first of all, we are extremely grateful for the valuable comments you have made on our manuscript. We have carefully read and considered your viewpoints. In response to the issues you mentioned, we have adjusted the text of the procedures, as shown in the highlighted part (Lines 269 to 272). Thank you again for your precious feedback.
Point 2: Can the authors also please clarify whether all children were tested within the same calendar year, or was this sample built up over multiple years of testing? With the broader context of the global COVID-19 pandemic, this kind of temporal information seems relevant for the reader to contextualize the findings.
Response 2: Dear Reviewer, first of all, we are extremely grateful for the valuable comments you have made on our manuscript. We have carefully read and considered your viewpoints. In response to the issues you mentioned, we have adjusted the text of the participants, as shown in the highlighted part (Lines 260 to 263). Thank you again for your precious feedback.
Point 3: In addition, as I reviewed the discussion, many of the phrases used suggested extreme cases of parental stress, problem behaviors or learning disorders. However, looking at the Tables provided, there is limited information about whether this sample adequately captured a population that was representative of these dire outcomes. Either additional information in the Instruments section about any relevant cut-off values for “severe” levels of each variable, or a figure near Table 2 that transparently displays the spread of values for each variable would add important context to the findings. While the Mean, SD, and ranges in Table 2 are very well presented and helpful, they do not tell the full story of who is included in this sample.
Response 3: Dear Reviewer, first of all, we are extremely grateful for the valuable comments you have made on our manuscript. We have carefully read and considered your viewpoints. In response to the issues you mentioned, we have adjusted the text of the instruments, as shown in the highlighted part (Lines 305 to 308, 321 to 323, 335 to 337). Thank you again for your precious feedback.
Point 4: line 106-107 “Rogers et al. (2009) found that high parental stress caused parents to control their children's academics more, leading to poor academic performance.” – The words “caused” and “leading to” in this sentence are inappropriate as the study cited is cross sectional which means it cannot inspect causation or longitudinal changes in academics. Cross-sectional studies can only describe associations between variables. I also believe it is important to note in the text that the sample in that particular study was specific to children with ADHD.
Response 4: Dear Reviewer, first of all, we are extremely grateful for the valuable comments you have made on our manuscript. We have carefully read and considered your viewpoints. In response to the issues you mentioned, we have adjusted the text as shown in the highlighted part (Lines 131 to 136). Thank you again for your precious feedback.。
Point 5: Lines 182-184 “the gene-socioeconomic status interaction hypothesis of cognition and learning proposes that family socioeconomic status influences children's behavior and learning development (González et al., 2024; Tucker-Drob, 2009).” This description of the gene-socioeconomic status interaction hypothesis is leaving out a lot of critical context. If adding additional context about the gene-socioeconomic status interaction hypothesis is not a priority, the subsequent sentences in that paragraph are much stronger and may be sufficient.
Response 5: Dear Reviewer, first of all, we are extremely grateful for the valuable comments you have made on our manuscript. We have carefully read and considered your viewpoints. In response to the issues you mentioned, we have adjusted the text as shown in the highlighted part (Lines 221 to 226). Thank you again for your precious feedback.
Point 6: Lines 236-238 “Considering the intersection of the difficult child dimension of the Parental Stress Index-Short Form (PSI-SF) developed by Abidin (1990) with the problem behavior subsequently measured in this study.” – This sentence does not make sense and should be revised.
Response 6: Dear Reviewer, first of all, we are extremely grateful for the valuable comments you have made on our manuscript. We have carefully read and considered your viewpoints. In response to the issues you mentioned, we have adjusted the text as shown in the highlighted part (Lines 294 to 297). Thank you again for your precious feedback.
Round 2
Reviewer 1 Report
Comments and Suggestions for Authors
Dear authors, thank you for the work done on your revision, which has significantly improved the manuscript. However, I still have a few concerns.
My main concern is a potential general overinterpretation of the results due to the indirect effect size. Please take into consideration that very small effect sizes indicate little practical relevance (even if they are statistically significant). You should add this into your limitations section and generally use a more cautious phrasing when describing and interpreting the results related to the small effect. At 2.2.% variance explanation of the model, it is clear that other factors would have been more relevant for the relationship you seek to explore. You could state this and hypothesize which factors these could have been (e.g., based on literature) and suggest a deeper look into this for future research.
My second concern is accuracy when describing other studies. I checked one of your newly added sources and not only was its design incorrectly described but also the results/ aim of the cited work. This might be a single case but I urge you to check your other sources with adequate scientific accuracy, too.
Minor points:
ll. 82 f.: "This lack of a mechanism limits our understanding of the causal pathway and intervention nodes between the two" => should be "this lack of knowledge" or similar.
l. 148: please delete "longitudinal" and rephrase. The study you cited seems to be a cross-sectional investigation of post-pandemic parenting stress levels and states that these levels are at least as high or higher compared to during-pancemic levels. It is not a longitudinal study and it does not compare to pre-pandemic levels.
l. 152: should be importance not "im-portance"
l. 266: I would rather speak of "child sex" than "child gender", as these children are still very young.
l. 297: Should be "also,..." not "as well as"
ll. 305 ff.: The Parenting Stress Index has three validated cut-offs (at least in the English and German Version) that are based on T-values - not stressed, stressed and highly stressed. Is this not the case for the Chinese version? Also, I did not quite understand if you used only certain parts of the questionnaire or the full questionnaire for your study. If you only used parts, please also describe how many items the full questionnaire consists of and why you only chose certain items.
ll. 521 ff.: "Although the value of the indirect effect is relatively small (Fritz & MacKinnon, 2007), its statistical significance suggests that this effect path has theoretical value and practical significance". Please rephrase. Statistical significance is a means to establish systematic differences which are not based on coincidence. It is not a means to determine clinical relevance. This is done via the corresponding effect size. A very small statistically significant effect is still a very small effect, hence its clinical/ practical relevance is not be overinterpreted.
Author Response
Author response to report 2:
Point 1: My main concern is a potential general overinterpretation of the results due to the indirect effect size. Please take into consideration that very small effect sizes indicate little practical relevance (even if they are statistically significant). You should add this into your limitations section and generally use a more cautious phrasing when describing and interpreting the results related to the small effect. At 2.2.% variance explanation of the model, it is clear that other factors would have been more relevant for the relationship you seek to explore. You could state this and hypothesize which factors these could have been (e.g., based on literature) and suggest a deeper look into this for future research.
Response 1:
Dear Reviewer,
Thank you very much for the valuable comments on our manuscript. Regarding the issues you mentioned, first, we have supplemented this point in the specific content of the Discussion section. Second, we have added corresponding statements in the Limitations section, and the specific content is shown in the green-highlighted part (Lines 648 to 655). Thank you again for your precious comments.
Point 2: My second concern is accuracy when describing other studies. I checked one of your newly added sources and not only was its design incorrectly described but also the results/ aim of the cited work. This might be a single case but I urge you to check your other sources with adequate scientific accuracy, too.
Response 2: Dear reviewer, first of all, thank you very much for your valuable comments on our manuscript. We have also carefully read and thought about your opinions. We attach great importance to the issue you raised. Based on all the issues that need to be improved, we have reviewed the relevant literature cited in the manuscript again and confirmed the correct expression of the literature opinions in the text. For the issues that need to be improved, please see the response to each point below for details.
Point 2.1: 82 f.: "This lack of a mechanism limits our understanding of the causal pathway and intervention nodes between the two" => should be "this lack of knowledge" or similar.
Response 2.1:
Dear Reviewer,
First of all, we sincerely appreciate the valuable comments you provided on our manuscript and have carefully read and considered your insights. In response to the issue you mentioned, we have adjusted the wording in this section, as shown in the green-highlighted part (Lines 82 to 83).
Thank you again for your precious feedback.
Point 2.2: 148: please delete "longitudinal" and rephrase. The study you cited seems to be a cross-sectional investigation of post-pandemic parenting stress levels and states that these levels are at least as high or higher compared to pandemic levels. It is not a longitudinal study and it does not compare to pre-pandemic levels.
Response 2.2:
Dear Reviewer,
First and foremost, we sincerely appreciate the valuable comments you provided on our manuscript and have carefully read and reflected on your insights. Regarding the issue you mentioned, we re-read the specific content of the cited literature. We apologize for this error that occurred during the revision process. We have re-consulted relevant literature for this perspective and adjusted the text wording as well as the cited references, as shown in the green-highlighted part (Lines 147 to 154). Meanwhile, we have re-reviewed the literature cited for other viewpoints in the manuscript to ensure no similar issues exist. Thank you again for your precious feedback.
Point 2.3: 152: should be importance not "im-portance"
Response 2.3: Dear Reviewer, first of all, we sincerely appreciate the valuable comments you provided on our manuscript. In response to the issue you mentioned, we have adjusted the text in this section, as shown in the green-highlighted part (Line 155). Thank you again for your precious feedback.
Point 2.4: 266: I would rather speak of "child sex" than "child gender", as these children are still very young.
Response 2.4: Dear reviewer, first of all, thank you very much for your valuable comments on our manuscript. We have carefully read and thought about your views. In response to your question, we have adjusted the text in that place, see the green highlight (Line 270). At the same time, we have also adjusted this statement in other parts of the manuscript. Thank you again for your valuable comments.
Point 2.5: 297: Should be "also,..." not "as well as"
Response 2.5: Dear reviewer, first of all, thank you very much for your valuable comments on our manuscript. In response to your question, we have adjusted the text in that place, see the green highlighted part (Line 303). Thank you again for your valuable comments.
Point 2.6: 305 ff.: The Parenting Stress Index has three validated cut-offs (at least in the English and German Version) that are based on T-values - not stressed, stressed and highly stressed. Is this not the case for the Chinese version? Also, I did not quite understand if you used only certain parts of the questionnaire or the full questionnaire for your study. If you only used parts, please also describe how many items the full questionnaire consists of and why you only chose certain items.
Response 2.6:
Dear Reviewer,
First, we sincerely thank you for the valuable comments on our manuscript. Regarding your question, first, it should be clarified that we only used the parental distress and parent-child dysfunctional interaction dimensions in PSI-SF as the measurement indicators for parenting stress in this study, as shown in the green-highlighted part (Lines 305 to 308). Second, there are two reasons for our implementation: (1) To minimize construct overlap and ensure the discriminant validity between measurement tools, this study only used two subscales of PSI-SF, "parental distress (PD)" and "parent-child dysfunctional interaction (PCDI)". The "Difficult Child (DC)" dimension was intentionally excluded because it is conceptually highly similar to the child behavioral problems (such as conduct problems and hyperactivity/inattention) measured by SDQ. Meanwhile, existing studies have pointed out that using related scales without addressing construct overlap may lead to the risks of shared method variance and measurement redundancy (Farrell, 2010; Podsakoff et al., 2003), which is the biggest reason for this study's implementation, as shown in the green-highlighted part (Lines 300 to 303); (2) At the same time, we also consulted relevant studies on PSI-SF and found that whether only two factors of "parental distress (PD)" and "parent-child dysfunctional interaction (PCDI)" can still be used as a means to measure parenting stress. Empirical studies by scholars also support the two-factor structure of "parental distress (PD)" and "parent-child dysfunctional interaction (PCDI)" in PSI-SF, as shown in the green-highlighted part (Lines 303 to 305). Based on the above two reasons, this study was conducted in this way.
Regarding the critical value of parenting stress you mentioned, relevant studies in the Chinese context only used percentile ranks for the critical value of parenting stress. The following are the source of this literature and the original sentence in the text: "Mothers with a total score equal to or higher than the 90th percentile and lower than the 15th percentile were considered as high-stress and low-stress groups, respectively (Geng et al., 2008)." as shown in the green-highlighted part (Line 315).
Thank you again for your valuable comments.
Farrell, A. M. (2010). Insufficient discriminant validity: A comment on Bove, Pervan, Beatty, and Shiu (2009). Journal of business research, 63(3), 324-327. https://doi.org/https://doi.org/10.1016/j.jbusres.2009.05.003
Geng, L., Ke, X., Xue, Q., Chi, X., Jia, J., Chen, P., & Lu, Z. (2008). Maternal parenting stress and related factors in mothers of 6-month infants. Chin Pediatr Integr Tradit West Med, 27(6), 457-459. https://doi.org/CNKI:SUN:ZYEK.0.2008-06-002
Podsakoff, P. M., MacKenzie, S. B., Lee, J.-Y., & Podsakoff, N. P. (2003). Common method biases in behavioral research: a critical review of the literature and recommended remedies. Journal of applied psychology, 88(5), 879. https://doi.org/10.1037/0021-9010.88.5.879
Point 2.7: 521 ff.: "Although the value of the indirect effect is relatively small (Fritz & MacKinnon, 2007), its statistical significance suggests that this effect path has theoretical value and practical significance". Please rephrase. Statistical significance is a means to establish systematic differences which are not based on coincidence. It is not a means to determine clinical relevance. This is done via the corresponding effect size. A very small statistically significant effect is still a very small effect, hence its clinical/ practical relevance is not be overinterpreted.
Response 2.7: Dear reviewer, first of all, thank you very much for your valuable comments on our manuscript. In response to your question, we have reread the content and adjusted the text, and added literature support for the corresponding viewpoints, see the highlighted part (Lines 526 to 529). Thank you again for your valuable comments.
Reviewer 2 Report
Comments and Suggestions for Authors
Please indicate in the discussion section that : further studies should conduct follow-up research in attempt to incorporate more objective indicators of children's behaviors and academic achievements.
Author Response
Point 1: Please indicate in the discussion section that: further studies should conduct follow-up research in attempt to incorporate more objective indicators of children's behaviors and academic achievements.
Response 1:
Dear Reviewer,
First and foremost, we sincerely appreciate the valuable comments you provided on our manuscript and have carefully read and considered your insights. In response to the issue you mentioned, we have adjusted the text in the Discussion section, as shown in the blue-highlighted part (Lines 534 to 537). Thank you again for your precious feedback.
Round 3
Reviewer 1 Report
Comments and Suggestions for Authors
I do not have any further comments and would suggest to accept the paper for publication.